

# Spatiotemporal prediction of alpine wetlands under multi-climate scenarios in the west of Sichuan, China

Haijun Wang[1,2,*], Xiangdong Kong[1,3,*], Onanong Phewnil[2], Ji Luo[1], Pengju Li[1], Xiyong Chen[1] and Tianhui Xie[1]

[1] Research Center of Agricultural Economics, School of Economics, Sichuan University of Science and Engineering, Zigong, Sichuan, China
[2] Faculty of Environment, Kasetsart University, Bangkok, Thailand
[3] College of Management Science, Sichuan University of Science and Engineering, Zigong, Sichuan, China
* These authors contributed equally to this work.

Corresponding authors
Haijun Wang,
wanghaijun@suse.edu.cn
Xiangdong Kong,
kongxiangdong@suse.edu.cn

## ABSTRACT

**Background:** The alpine wetlands in western Sichuan are distributed along the eastern section of the Qinghai-Tibet Plateau (QTP), where the ecological environment is fragile and highly sensitive to global climate change. These wetlands are already experiencing severe ecological and environmental issues, such as drought, retrogressive succession, and desertification. However, due to the limitations of computational models, previous studies have been unable to adequately understand the spatiotemporal change trends of these alpine wetlands.
**Methods:** We employed a large sample and composite supervised classification algorithms to classify alpine wetlands and generate wetland maps, based on the Google Earth Engine cloud computing platform. The thematic maps were then grid-sampled for predictive modeling of future wetland changes. Four species distribution models (SDMs), BIOCLIM, DOMAIN, MAXENT, and GARP were innovatively introduced. Using the WorldClim dataset as environmental variables, we predicted the future distribution of wetlands in western Sichuan under multiple climate scenarios.
**Results:** The Kappa coefficients for Landsat 8 and Sentinel 2 were 0.89 and 0.91, respectively. Among the four SDMs, MAXENT achieved a higher accuracy ($\alpha = 91.6\%$) for the actual wetland compared to the thematic overlay analysis. The area under the curve (AUC) of the MAXENT model simulations for wetland spatial distribution were all greater than 0.80. This suggests that incorporating the SDM model into land change simulations has high generalizability and significant advantages on a large scale. Furthermore, simulation results reveal that between 2021 and 2100 years, with increasing emission concentrations, highly suitable areas for wetland development exhibit significant spatial differentiation. In particular, wetland areas in high-altitude regions are expected to increase, while low-altitude regions will markedly shrink. The changes in the future spatial distribution of wetlands show a high level of consistency with historical climate changes, with warming being the main driving force behind the spatiotemporal changes in alpine wetlands in western Sichuan, especially evident in the central high-altitude and northern low-altitude areas.

# INTRODUCTION

The western Sichuan area has typical alpine meadows and the unique alpine wetlands of the Qinghai-Tibet Plateau (QTP) (*Li et al., 2024a*). These wetlands include Zoige, which is China's largest marsh wetland. The distinctive environment fosters a rich diversity of flora and fauna, serving as the last habitat for several endangered species such as alpine water parsley, bulrushes, and the black-necked crane (*Scott, 1993*; *Li et al., 2024b*). Additionally, being located in a region sensitive to global climate change and ecological fragility (*Wang et al., 2024*), it is highly susceptible to external disturbances. Therefore, studying the spatiotemporal changes in wetlands in this area is crucial for regional biodiversity conservation, economic development in ethnic regions, and achieving carbon peak and carbon neutrality goals. Furthermore, from the perspective of national ecological resource protection, the alpine wetlands in western Sichuan play a significant role in establishing an ecological safety barrier in the western part of the country.

Previous studies on spatiotemporal changes in wetlands have primarily relied on wetland area transition matrices and driving factors (*Anand & Oinam, 2020*; *Ansari & Golabi, 2019*; *Peng et al., 2020*), with commonly used predictive models including CA-Markov and CLUE-S. These models obtain land cover change data through multi-period remote sensing images and integrate driving factors for prediction (*Xu et al., 2023*). Although these models perform excellently in land cover change studies at smaller scales, their application at larger scales (*e.g.*, provincial, national, or intercontinental) is limited, primarily due to inefficiencies in obtaining time-series remote sensing classification data (*Xu et al., 2022*) and the complexity of constructing driving factor systems (*Peng et al., 2020*). Existing research shows that factors influencing species habitat distribution include climate and topography (*Pecl et al., 2017*; *Schneider et al., 2020*), while the natural driving factors for wetland changes are mainly temperature, precipitation, and topography (*Anand & Oinam, 2020*; *Ansari & Golabi, 2019*; *Peng et al., 2020*). Thus, species and wetland spatial distributions have similar environmental variable characteristics. Consequently, applying species distribution theory to study spatiotemporal changes in wetlands presents a potentially feasible interdisciplinary approach. There are precedents for the successful application of ecological models in other fields; for example, the Minimum Cumulative Resistance (MCR) model, originally from landscape ecology, has been successfully introduced into urban planning (*Wang et al., 2019a*). Currently, widely used SDM models include MAXENT (*Philips, 2009*), GARP (*Stockwell, 1999*), BIOCLIM (*Busby, 1991*), and DOMAIN (*Belbin, 1992*), *etc.*

In this context, we employed the Google Earth Engine (GEE) computing platform and high-resolution remote sensing imagery to classify alpine wetlands in western Sichuan. Building on this classification, we innovatively introduced the species distribution model (SDM) and applied Coupled Model Intercomparison Project Phase 6 (CMIP6) climate data to predict the spatial changes of these wetlands across four time periods: 2021–2041,

2041–2061, 2061–2081, and 2081–2100. This study aims to explore future trends in alpine wetland changes and, in light of the ecological challenges they face, propose targeted conservation strategies.

## MATERIALS AND METHODS

### Study area

The western Sichuan (WS) region (Fig. 1) is located in the eastern section of the Qinghai-Tibet Plateau and the Hengduan Mountains. It mainly comprises the Ganzi-Aba Plateau and the western Sichuan mountainous areas, with an average altitude of approximately 4,000–4,500 m above sea level. The climate is influenced by both continental and monsoon climates, with annual precipitation ranging from 600 to 1,000 mm. The region features a dense river network, including major water systems such as the Yellow River, Jinsha River, Yalong River, Dadu River, and Min River. It includes the largest alpine mire belt in the south (Zoige-Hongyuan-Aba region) and numerous plateau lakes (Little Sea), forming extensive alpine lake wetlands. Western Sichuan has 11 wetland ecological protection zones, of which three are national-level and four are provincial-level. Wetlands are primarily distributed around alpine meadows and lakes, serving as habitats for rare and endangered species and forming an important section of the national key ecological function zones. The region's vegetation is predominantly alpine meadows with significant ecological and environmental value.

## DATASET

In this study, the images required for wetland classification include 40 scenes of Sentinel 2 and 25 scenes of Landsat 8, covering 2021 to 2024 years. These images are obtained from the online database of the GEE platform (https://developers.google.com). Environmental variable data, including air temperature and precipitation, are in numerical format and sourced from the National Meteorological Science Data Center (https://data.cma.cn/), covering 12 meteorological stations within the study area. The raw meteorological data were processed using spatial interpolation to produce air temperature and precipitation raster data with a spatial resolution of 30 s. Terrain data utilized in this study are derived from the ASTER DEM with a spatial resolution of 30 m, downloaded from GEE. The future climate data used are simulated based on the CMIP6 framework under Shared Socioeconomic Pathways (SSP) scenarios. Climate data from three scenarios, SSP1 (Sustainable Development), SSP2 (Middle Path), and SSP5 (Fossil Fuel Driven) are selected. These data include air temperature, precipitation, and bioclimatic raster layers with a spatial resolution of 30 s, sourced from WorldClim (https://www.worldclim.org).

## METHODS

### Classified sample

Wetland distribution samples are crucial for predicting future wetland spatial changes. Therefore, we classified alpine wetlands using remote sensing imagery to extract their spatial distribution, which serves as the basis for predicting future wetland changes. GEE platform enables more efficient large-scale remote sensing image classification (*Wang et al., 2022*).
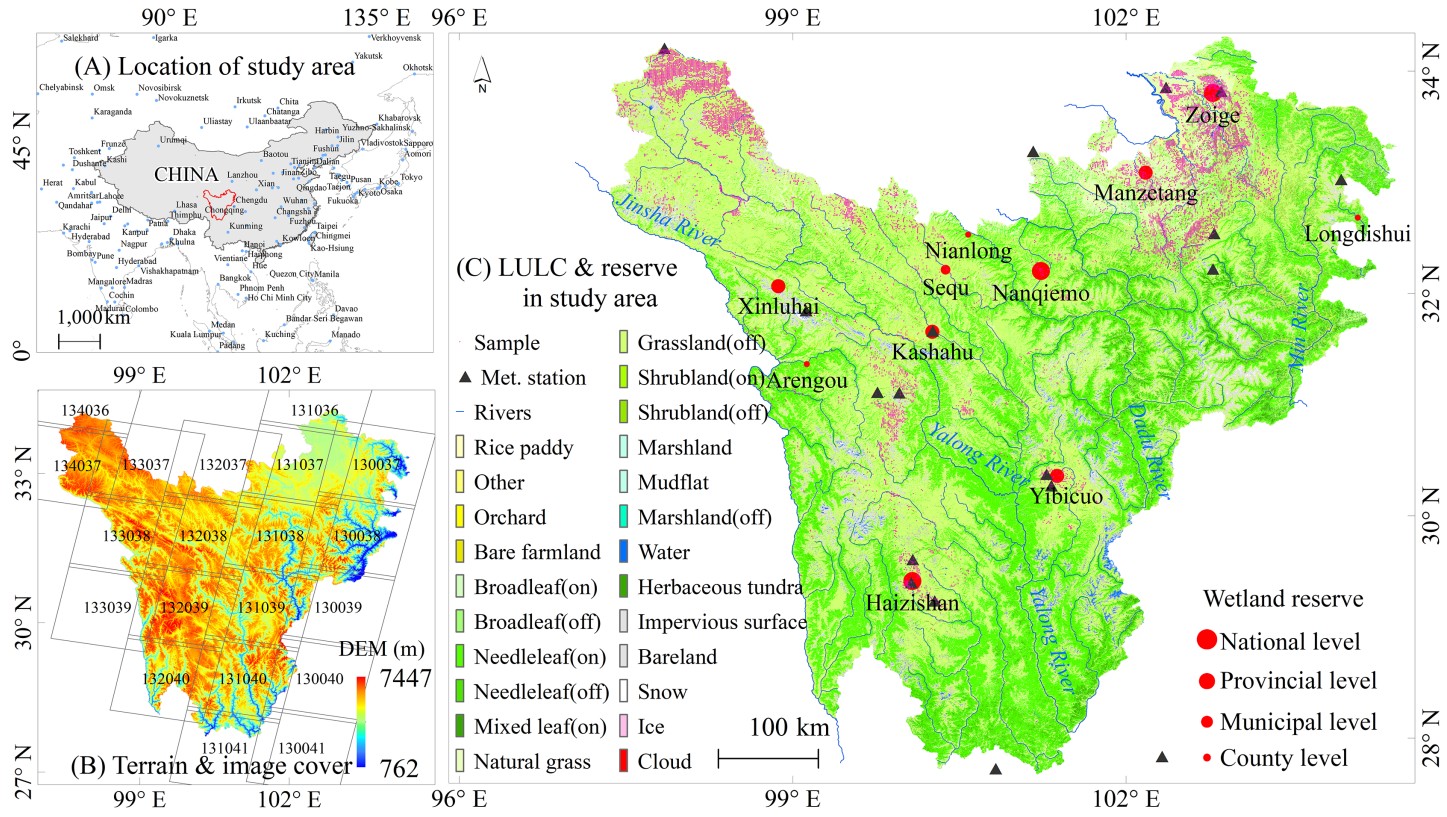

**Figure 1  Geo-spatial features of western Sichuan.** (A), (B), and (C) are location, terrain & image cover, and land use, respectively.

Thus, we utilized the GEE platform to obtain 40 Sentinel 2 images and 25 Landsat 8 images from July to August 2021–2024 for the western Sichuan region. Given the low distinction between marsh wetlands and alpine meadows, we used high-resolution satellite data (Cbers-04 with 5 m resolution), Google Maps, and 30 m land cover data to aid in sample selection. In total, we collected 2,100 wetland samples in the study area (70% for classification and 30% for validation). The GEE platform offers various classifiers. We selected four classification algorithms: Classification and Regression Trees (CART), support vector machines (SVM), random forest (RF), and Bayesian classification (Bayes). These algorithms were used to classify Sentinel 2 and Landsat 8 images, producing wetland thematic data with 10 and 15 m resolutions. This thematic data was used as the prediction sample for wetland spatial distribution. The GEE classification process is shown in Fig. 2.

## Defining environmental variables

Environmental variables are factors driving wetland changes. This study defines two sets of environmental variables for simulating both the current state of wetlands and their future distribution. (1) Current environmental variables: We used raw meteorological data from 2021 to 2024 to derive monthly maximum temperature (TX), monthly minimum temperature (TN), and monthly total precipitation (PR) from 12 meteorological stations in the western Sichuan region. Kriging interpolation (*Matheron, 1963*) was applied using

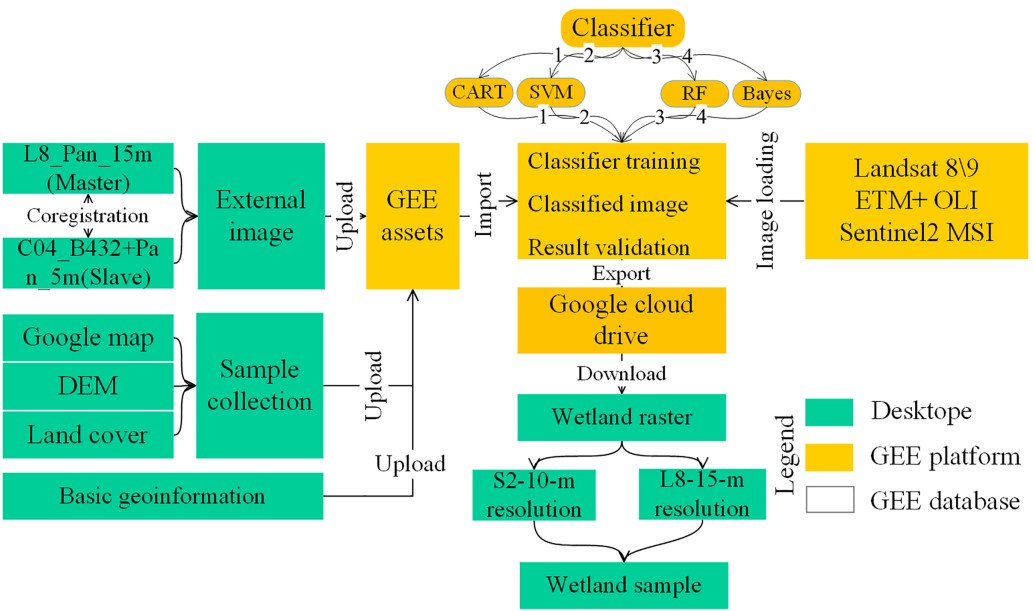

**Figure 2 Wetland remote sensing classification workflow in GEE.**

ArcGIS 10 software to produce spatial grid data with a resolution of 30 s. We corrected the temperature raster using the DEM, based on a temperature decrease of 0.65 °C for every 100-m increase in elevation. Slope and aspect information was extracted from the ASTER DEM and resampled to obtain elevation (H), slope (SLO), and aspect (ASP) data, also at a 30-s resolution. Due to the sparsely populated nature of the region and the minimal impact of human activities on wetland development, population density, and socio-economic factors were excluded from the environmental dataset. (2) Future environmental variables: For simulating future wetland distribution, we used environmental variables from the WorldClim future climate dataset. This includes TN, TX, PR, bioclimatic variables (BC), and terrain information such as H, SLO, and ASP.

## Wetland prediction

A big wetland sample set (10,931 samples) based on gridded wetland thematic map and environmental datasets (TN, TX, PR, BC, H, SLO, and ASP) was employed to simulate the current distribution of alpine wetlands in western Sichuan using four models: BIOCLIM, DOMAIN, MAXENT, and GARP (Table 1). The 10 and 15 m resolution wetland thematic map was overlaid with the simulation results, and the optimal model was selected by comparing the overlap area ratio ($\alpha$).

We selected SSP1 (2.6 W/m$^2$, sustainable), SSP2 (4.5 W/m$^2$, intermediate), and SSP5 (8.5 W/m$^2$, fossil-fuel) under the Shared Socioeconomic Pathways (SSPs) scenario. MAXENT was employed as the primary model, with BIOCLIM, DOMAIN, and GARP (Table 2) utilized as auxiliary models to predict the future spatial distribution trends (July–August) of alpine wetlands in western Sichuan.

**Table 1 Principles of four SDMs and their implementation tools.**

| Model | Definition | | Software |
|---|---|---|---|
| BIOCLIM | $S_i = \prod_{j=1}^{n} \left( \dfrac{x_j - x_{j,min}}{x_{j,max} - x_{j,min}} \right)$ | Where $S_i$ is the suitability index for point $i$, $n$ is the number of climatic variables, $x_j$ is the value of climatic variable $j$ at point $i$, an $x_{j,max}$ and $x_{j,min}$ are the upper and lower bounds of climatic variable $j$, respectively. | DIVA-GIS |
| DOMAIN | $S_i = \dfrac{1}{\sum_{j=1}^{n} d(x_i, x_j)}$ | Where $S_i$ is the suitability score of target point $x_i$, $d(x_i, x_j)$ is the environmental distance between target point $x_i$ and consistency distribution point $x_j$, and $n$ is the number of known distribution points. | DIVA-GIS |
| MAXENT | $\hat{p}(x) = \dfrac{1}{z} \exp \left( \sum_{j=1}^{m} \lambda_j f_j(x) \right)$ | Where $\hat{p}(x)$ is the species distribution probability under environmental variable $x$, $z$ is the normalization constant, $\lambda_j$ are model parameters, $f_j(x)$ is the characteristic function of environmental variable $x$, and $m$ is the number of characteristic functions. | MAXENT |
| GARP | $F(x) = \sum_{i=1}^{k} w_i R_i(x)$ | Where $F(x)$ is the suitability score of target point $x$, $R_i(x)$ is the suitability of rule $i$ at point $x$, $w_i$ is the weight of rule $R_i$, and $k$ is the number of rules. | GARP MS |

**Table 2 Environmental variables and models for predicting spatiotemporal changes under different periods and scenarios.**

| Period | Scenarios | | Variables | Resolution | Main model | Auxiliary model |
|---|---|---|---|---|---|---|
| 2021–2040 | SSP | 2.6 W/m$^2$ | SAM | 30″ | MAXENT | BIOCLIM |
| | | 4.5 W/m$^2$ | TN | | | DOMAIN |
| | | 8.5 W/m$^2$ | TX | | | GARP |
| 2041–2060 | | 2.6 W/m$^2$ | PR | | | |
| | | 4.5 W/m$^2$ | BC | | | |
| | | 8.5 W/m$^2$ | H | | | |
| 2061–2080 | | 2.6 W/m$^2$ | SLO | | | |
| | | 4.5 W/m$^2$ | ASP | | | |
| | | 8.5 W/m$^2$ | | | | |
| 2081–2100 | | 2.6 W/m$^2$ | | | | |
| | | 4.5 W/m$^2$ | | | | |
| | | 8.5 W/m$^2$ | | | | |

# RESULTS

## Alpine wetland classification

Supervised classifications of Landsat 8 and Sentinel 2 images were performed on the GEE platform, yielding Kappa coefficients of 0.89 and 0.91, respectively. The classification data were analyzed, and marsh wetlands were extracted, as shown in Fig. 3. The extraction results from the two images show that the total area of alpine marsh wetlands in western Sichuan is 17,438.97 and 17,454.56 km$^2$, with a difference of 15.60 km$^2$. The Sentinel 2 classification results indicate that marsh wetland resources are mainly distributed in Xinlong, Shiqu, Zoige, Hongyuan, and Ganzi, with reserves of 4,703.08, 2,757.97, 2,755.50, 2,066.52, and 1,614.14 km$^2$, respectively. There are fewer marsh wetland resources in Muli, Danba, Heishui, Lixian, Maerkang, and Mao County. Compared to the results of the second Sichuan wetland resource survey (*China National Forestry Bureau, 2015*), the total
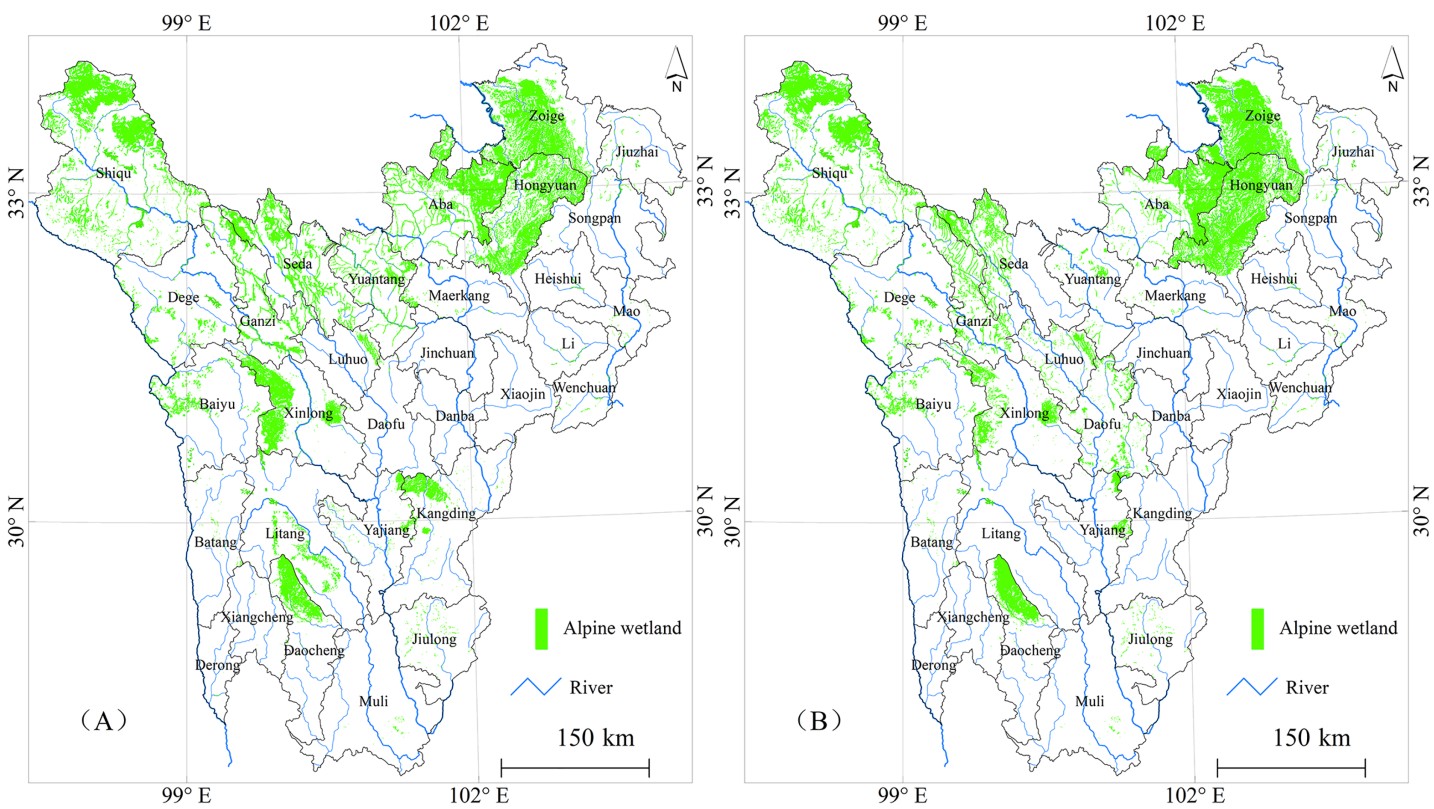

**Figure 3 Remote sensing classification and extraction results of alpine wetlands in Western Sichuan.** (A) Landsat 8–15 m image, (B) Sentinel 2–10 m image.

area of marsh wetlands extracted from Sentinel 2 images increased by 37.42 km². The increase is mainly observed in Ganzi, Xinlong, Aba, Rangtang, and Maerkang areas. Specifically, Maerkang experienced an increase of 0.81 km², mainly distributed in the border area with Aba. The second Sichuan wetland resource survey used the Landsat 8–30 m image, which did not allow for accurate classification and identification of small wetland patches, and thus, such wetlands were not included in the statistics. The higher resolution of the current images enables better identification and extraction of smaller marsh wetlands.

## Simulation and validation

Historical climate data combined with topographic data were used as environmental variables to simulate the current distribution of wetlands in western Sichuan. The simulation results were validated using thematic wetland data to evaluate the performance of the SDM model for assessing the suitability of wetland spatial distribution. The results of the four SDM models simulating the current distribution of alpine wetlands in western Sichuan are shown in Fig. 4. The simulation results of the four models divided the wetland spatial distribution in the study area into unsuitable areas (No suitable), low suitable areas (Low), medium suitable areas (Medium), high suitable areas (High), very high suitable areas (Very High), and excellent suitable areas (Excellent).

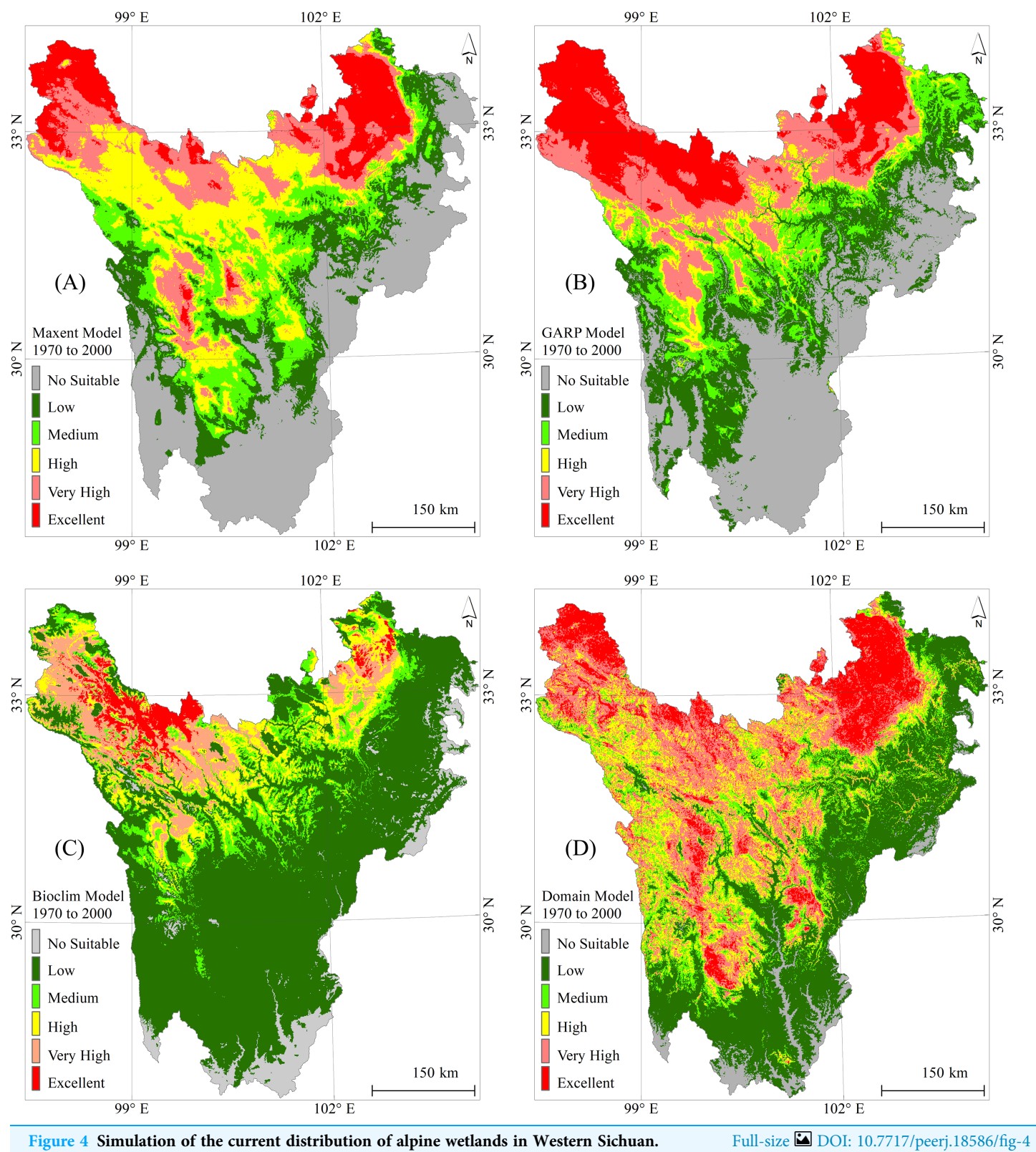

**Figure 4 Simulation of the current distribution of alpine wetlands in Western Sichuan.**

The simulation results of MAXENT and GARP are quite similar, with their classifications of unsuitable and low suitable areas closely aligned. The overlay analysis of MAXENT's excellent suitable area and the actual wetland distribution showed an α value of 91.6%, while GARP's excellent suitable area was slightly higher than the actual wetland distribution (α = 88.5%). In BIOCLIM's simulation results, the area of the extremely suitable wetland distribution was smaller than the actual wetland distribution (α = 79.5%), indicating poor simulation performance in high-altitude areas. In contrast, DOMAIN's simulation results showed a reasonable distribution of the extremely suitable wetland area, but its area was higher than the actual wetlands (α = 76.3%). From the α values comparing the simulation results of the four models with the actual wetlands, MAXENT had the best simulation performance, followed by GARP, and then BIOCLIM and DOMAIN. Therefore, MAXENT was selected as the main model, while the other three models participated in the simulation with their results used for reference and comparison only.

## Prediction of changes

Using the main model MAXENT and three auxiliary models, predictions were executed for the spatial distribution changes of alpine wetlands in western Sichuan during four periods from 2021 to 2100, as shown in Fig. 5. Predictions reveal that during the period from 2021 to 2040, under the low emission scenario (SSP2.6), the area of extremely suitable wetland distribution is higher compared to the high emission scenario (SSP8.5), particularly in the northern, lower elevation regions of the study area. As temperatures rise under SSP2.6 and SSP8.5 scenarios, MAXENT predictions indicate a reduction in the distribution area of very high suitability and an increase in high suitability. The results suggest a contraction of areas highly suitable for alpine wetland distribution, a trend also observed in BIOCLIM predictions. From the MAXENT model predictions, it is evident that during 2041–2060, with regional warming, the area of extremely suitable wetland distribution decreased, with a more pronounced reduction in very high suitability regions in Xinlong and Baiyu counties. GARP model results show a notable reduction in high-medium suitable wetland areas toward higher latitudes. The model predictions exhibit that increased carbon emissions led to a warming effect, which raises surface evaporation, negatively impacting alpine wetland development.

During 2061–2080, MAXENT predicts an increase in the area of high-suitability wetland distribution in high-altitude areas of Xinlong County and Haizishan. This increase is attributed to the higher temperature leading to increased ice and snow melt, which enhances the water supply to lakes and marshes in high-altitude areas. DOMAIN model results show no significant differences in alpine wetland distribution under the three emission scenarios. GARP model results indicate a reduction in areas of extremely suitable wetland distribution, while moderate suitability areas see a slight increase, primarily in the Minjiang and Dadu River valley regions. During 2081–2100, MAXENT predictions present that high carbon emission scenarios lead to warming in the central high-altitude areas of the study area, improving the suitability for wetland distribution.

By analyzing the MAXENT prediction results and comparing them with the other three prediction results, it was found that during the period from 2021 to 2100, under the low

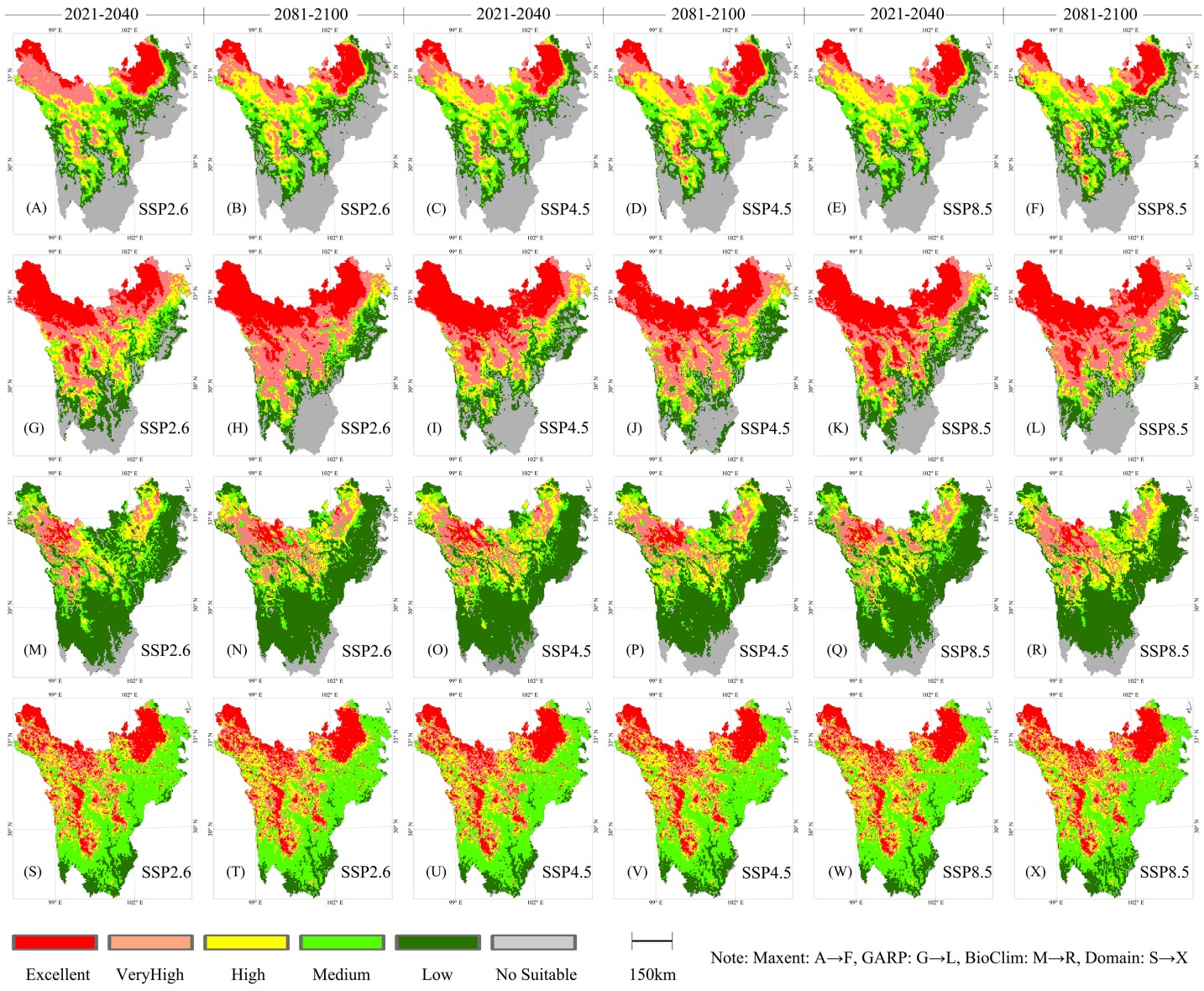

**Figure 5 Prediction of the spatial distribution of alpine wetlands from 2021 to 2100 in Western Sichuan.**

emission scenario (SSP2.6), the changes in the high suitability distribution area of alpine wetlands in western Sichuan are not significant. This is mainly because the temperature rise in western Sichuan is not significant under the low emission scenario, and the distribution range of alpine wetlands in western Sichuan is significantly affected by temperature. Under the intermediate emission scenario (SSP4.5), the suitability of wetland distribution in the central high-altitude areas of the study region increases. The increase in ice and snow melts due to warming enhances the water supply to these high-altitude regions, and the mid-latitude high-altitude areas of western Sichuan are more noticeably affected by the warming. Under the high emission scenario (SSP8.5), the area of extremely suitable wetland distribution in high-latitude areas decreases, while the area of extremely

suitable wetland distribution in central high-altitude areas shows an increasing trend. The high-latitude areas of alpine wetland distribution in western Sichuan are mainly located in Zoige and Shiqu counties, where warming increases surface evaporation. Additionally, the wetland water supply in this region mainly comes from surface runoff and precipitation, and increased evaporation affects the extent of wetland areas.

## DISCUSSION

### Results evaluation

The future spatial distribution of wetlands is influenced by various factors, leading to numerous uncertainties in distribution patterns. Therefore, we discuss the prediction accuracy by combining the ROC analysis of the prediction process with previous related studies. Referencing the methods of *Elith et al. (2011)* and *Merow, Smith & Silander (2013)* to evaluate the MAXENT prediction results, during the period from 2021 to 2040, the area of regions extremely suitable for alpine wetland distribution decreases with increasing emission concentrations across the three emission scenarios. The ROC analysis of the prediction results shows that the AUC values (Figs. 6A, 6B, and 6C) are 0.804, 0.809, and 0.804, respectively, all greater than 0.80 (*Muschelli, 2020*; *Huang & Ling, 2005*), demonstrating high prediction accuracy. The wetland prediction results for the period from 2041 to 2060 are similar to those from 2021 to 2040, with fewer regions of very high suitability for alpine wetlands in Xinlong and Baiyu counties. During the periods from 2061 to 2080 and 2081 to 2100, the warming of the central high-altitude areas of western Sichuan leads to improved suitability for wetland distribution. The AUC values for the prediction results during the period from 2081 to 2100 (Figs. 6D, 6E, and 6F) are 0.808, 0.809, and 0.812, respectively, all exceeding 0.80, confirming the reliability of the predictions.

We analyzed the historical changing trends in temperature and precipitation in western Sichuan (Figs. 7 and 8). Observational data from 12 meteorological stations indicate a rising temperature trend in western Sichuan, while precipitation shows fluctuating changes with significant spatial variations. The increase in temperature will have a substantial impact on the alpine wetlands of western Sichuan, particularly affecting surface runoff in high-altitude areas and evaporation in low-altitude areas. We also cross-analyzed recent studies on alpine wetlands in western Sichuan to verify the reliability of our results. This study predicts the spatial distribution of alpine wetlands in western Sichuan based on three emission scenarios. Under the high emission scenario, regional temperatures show a significant increasing trend, which will markedly impact wetland habitats. The warming leads to increased evaporation, causing the surface in low-altitude areas to become drier and damaging the wetland development environment. This is particularly evident in the north of the study area, such as Zoige, Ganzi, and Aba, consistent with previous studies (*Xue et al., 2014*; *Li et al., 2022*; *Yu et al., 2023*; *Cao et al., 2020*). For wetlands in high-altitude areas, warming has a positive effect on wetland development. For instance, the central part of the study area, including Litang County and Daocheng County, has higher elevations (Fig. 1B) and numerous marsh wetlands. In this region, the overall climate shows a trend of rising temperatures with no significant changes in precipitation (*Wang, He & Niu, 2020*; *Wang et al., 2022*; *Zhang et al., 2021*; *Wang et al., 2019b*), which

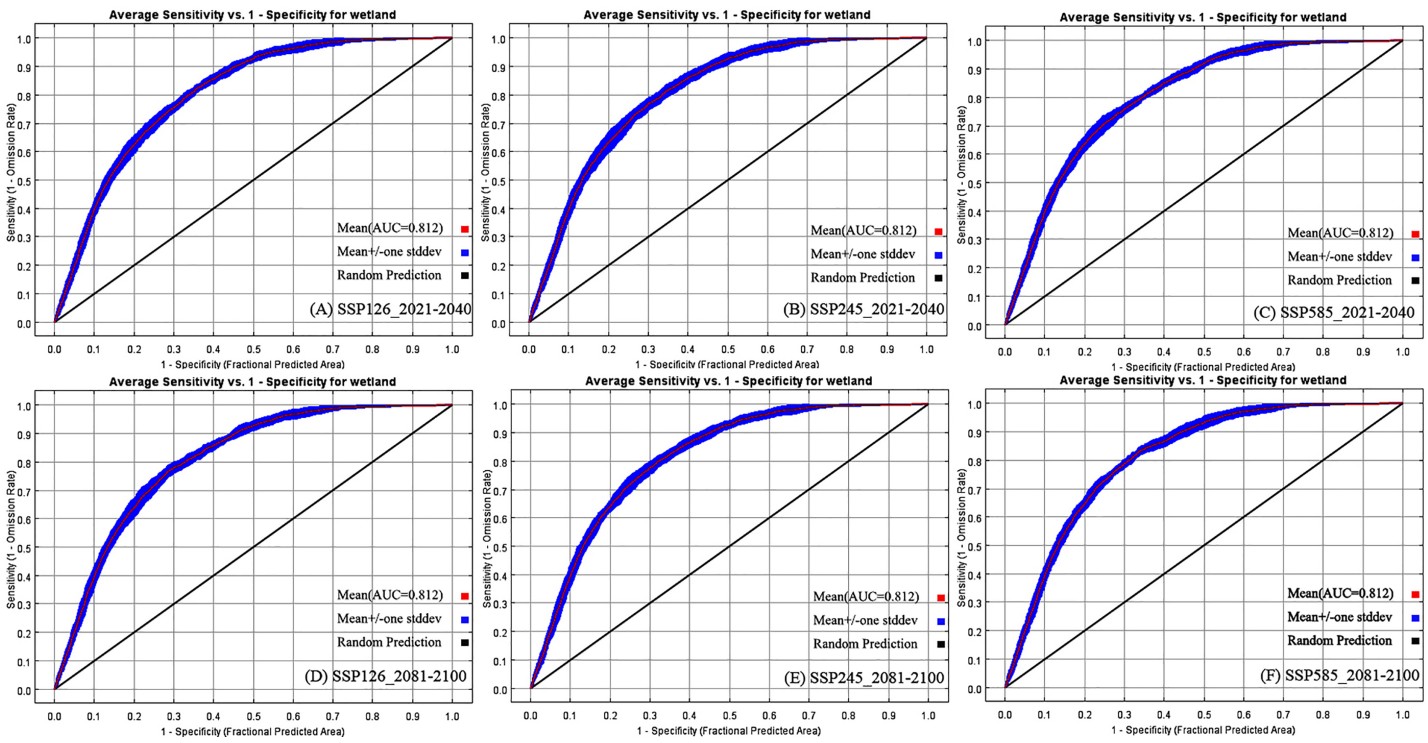

**Figure 6 ROC analysis of the alpine wetland prediction.**

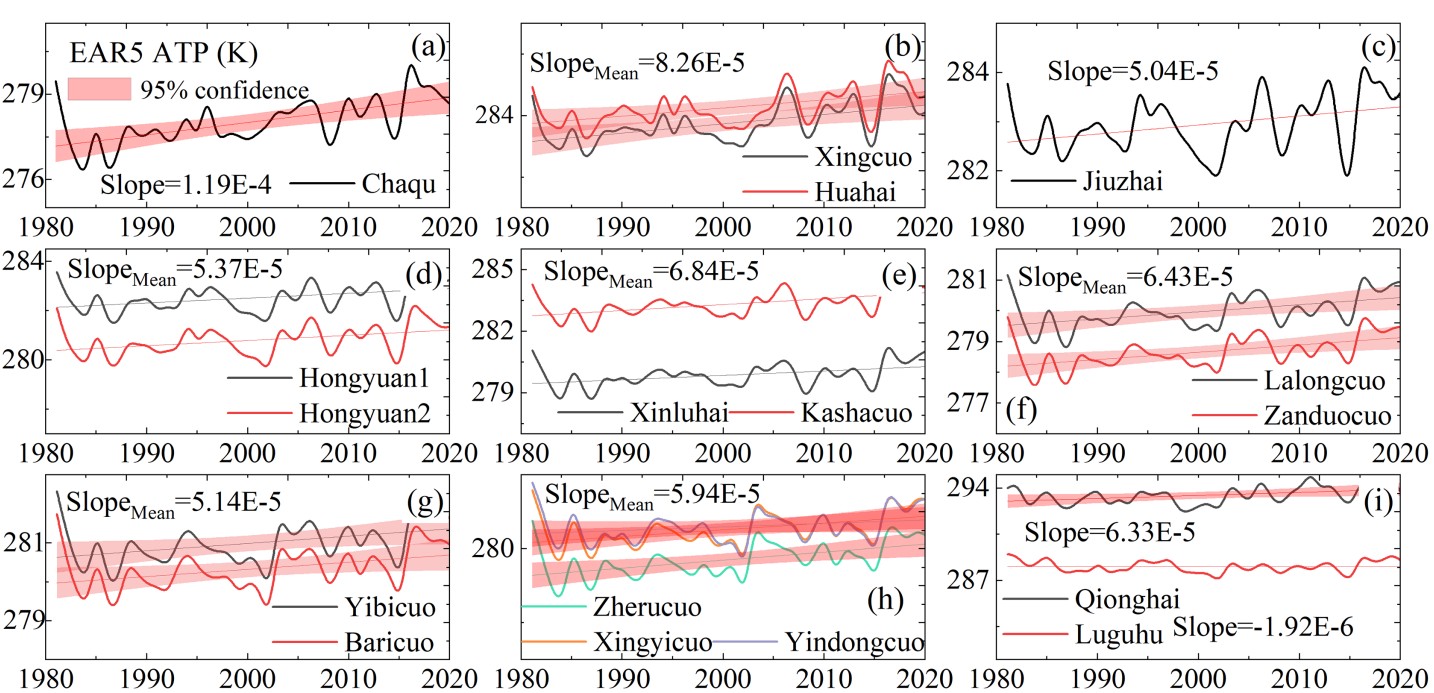

**Figure 7 ATP change trend of alpine wetlands at the site scale (1980–2020).** Latitude decreases from sub-figure (A) to (I). Sites (A), (D), (G), (F), and (H) are above 4,500 m, (B), (C), (E) range from 3,000 to 3,900 m, and (I) are below 3,000 m. (Note: $Slope_{Mean}$ represents the average slope, $Slope_{Up\ mean}$ indicates the average upward trend and $Slope_{Down\ mean}$ denotes the average downward trend, all within a sub-figure.).

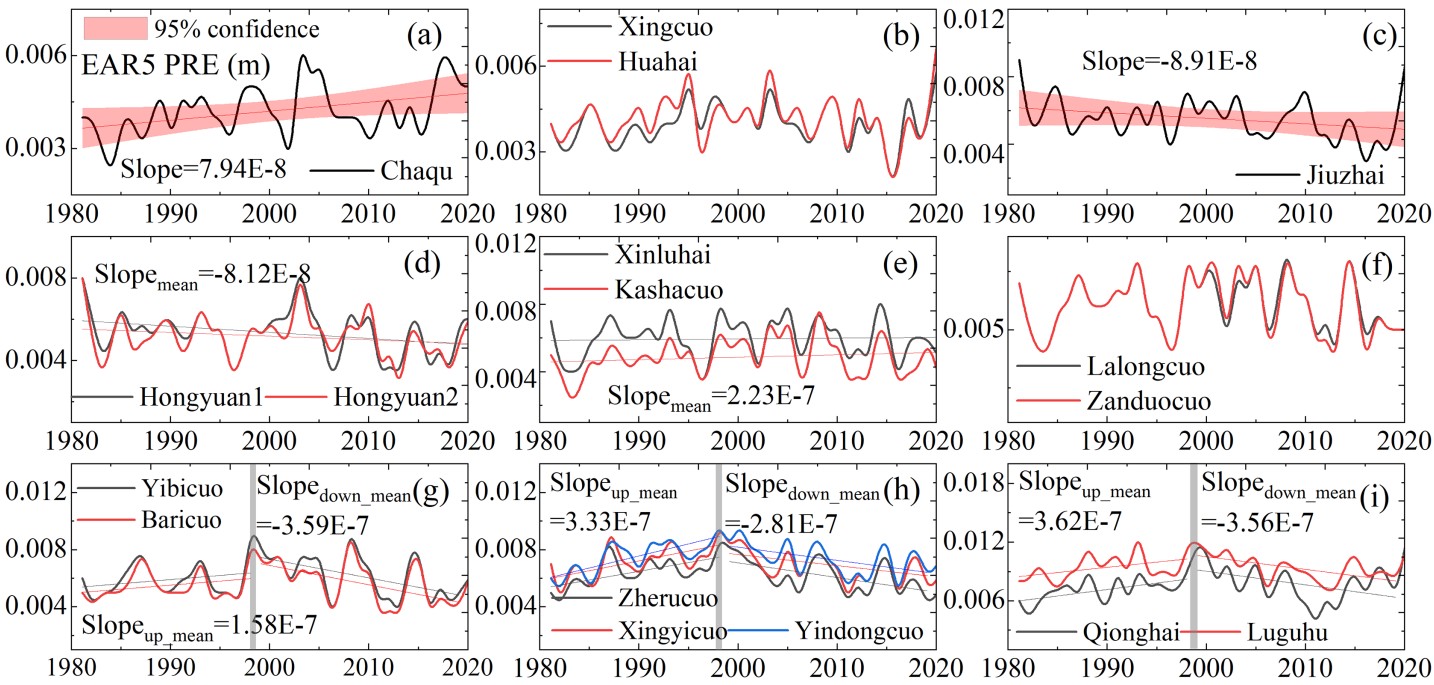

**Figure 8 PRE change trend of alpine wetlands at the site scale (1980–2020).** Latitude decreases from sub-figure (A) to (I). Sites (A), (D), (G), (F), and (H) are above 4,500 m, (B), (C), (E) range from 3,000 to 3,900 m, and (I) are below 3,000 m. (Note: $Slope_{Mean}$ represents the average slope, $Slope_{Up\ mean}$ indicates the average upward trend and $Slope_{Down\ mean}$ denotes the average downward trend, all within a sub-figure.).

accelerates the melting of mountain ice and snow, providing a favorable environment for wetland development. Our prediction results under the high emission scenario show an increase in the extremely suitable zone for wetland distribution in Litang County, the central study area, which aligns closely with the observed trends in temperature and precipitation changes.

## Limitation

This study primarily utilized seven factors: TN, TX, PR, BC, H, SLO, and ASP. The factors influencing changes in alpine wetlands are complex and diverse (*Wang et al., 2022*), encompassing various aspects of the natural environment and socio-economic conditions. Although the impact of socio-economic development on future climate change has been considered within the CMIP6 framework (*Juckes et al., 2020*), there is still a lack of consideration for aspects such as grazing and tourism development at the medium and small scales. Therefore, related research should focus on refining environmental variables and developing simulated data for future socio-economic development. In this context, enhancing the study of driving forces behind alpine wetland changes, alongside a thorough understanding of the natural environmental and socio-economic impacts, is essential for building a robust and precise evaluation system for alpine wetland dynamics.

Another issue is the ground validation of results regarding future changes in alpine wetlands, which remains a challenge as indicated by existing literature (*Philips, 2009*; *Elith et al., 2011*; *Merow, Smith & Silander, 2013*). Current studies mainly rely on the area under

the AUC curve (*Elith et al., 2011*; *Merow, Smith & Silander, 2013*) or use cross-validation with results from other models (*Yang et al., 2022*; *Moradi, Ashrafzadeh & Naghipour, 2024*). This study has assessed the Maxent simulation results using AUC and performed cross-validation on Maxent by integrating the predictions from GARP, BIOCLIM, and DOMAIN. Therefore, establishing an efficient automated ground monitoring system to collect environmental change data from typical alpine wetland distribution areas in western regions is of great value for future ground validation of alpine wetland change predictions.

## CONCLUSIONS

Using Landsat 8 (15 m) and Sentinel 2 (10 m) images for alpine wetlands as samples, and selecting key factors driving changes in alpine wetlands as environmental variables, we utilized four SDM models to predict the spatiotemporal trends of alpine wetlands in western Sichuan under three SSP emission scenarios. The Kappa coefficients for Landsat 8 and Sentinel 2 were 0.89 and 0.91, respectively. Among the four SDMs, MAXENT achieved a higher accuracy ($\alpha$ = 91.6%) for the actual wetland compared to the thematic overlay analysis. The prediction results show that the AUC values for all three emission scenarios are greater than 0.8, suggesting that the models perform well and are suitable for predicting the spatial distribution of alpine wetlands at a large scale using SDM models. Moreover, the predictions reveal that from 2021 to 2100, under the low emission scenario, the area of the high suitability zones for alpine wetlands in western Sichuan did not change significantly. Under the medium emission scenario, the suitability of wetland distribution in the high-altitude areas of the central study region increases. Under the high emission scenario, the area of extremely suitable zones for wetland distribution in high-latitude regions increases notably. Based on the predicted results for alpine wetlands and considering the geographical and environmental characteristics of western Sichuan, we propose four ecological protection strategies: establishing a wetland resource data-sharing platform, improving the wetland monitoring system, balancing wetland conservation with economic development, and enhancing research on the driving mechanisms of wetland changes.

## ACKNOWLEDGEMENTS

We would to thank GEE, NMSDC, and WorldClim for providing computational services and data support for this research, as well as the anonymous reviewers for their constructive and detailed comments.

### Funding

This work was supported by the Yangtze River Key Ecological Functional Area Protection Policy Research Center Project (YREPC2024-ZD001), the Southern Sichuan Development Research Institute of the Chengdu-Chongqing Economic Circle Project (CYQCNY20242), the Opening Fund of the Sichuan Key Provincial Research Base of Intelligent Tourism Project (ZHZR23-01, ZHZR24-03), the Social Development and Social Risk Control

Research Center Project (SR24A14), the Rural Community Governance Research Center Project (SQZL2024A01), the Agricultural Modernization and Rural Revitalization Research Center Project (AMRR2024001), the Key Laboratory of Digital-Intelligent Management and Ecological Decision Optimization for Liquor in the Upper Yangtze River Region Project (ZDSYS24-05), Western Ecological Civilization Research Center Project (XBST2024-ZC003), and the System Science and Enterprise Development Research Center Project (XQ23B03). The funders had no role in study design, data collection and analysis, decision to publish, or preparation of the manuscript.

## Grant Disclosures

The following grant information was disclosed by the authors:

Yangtze River Key Ecological Functional Area Protection Policy Research Center Project: YREPC2024-ZD001.

Southern Sichuan Development Research Institute of the Chengdu-Chongqing Economic Circle Project: CYQCNY20242.

Sichuan Key Provincial Research Base of Intelligent Tourism Project: ZHZR23-01, ZHZR24-03.

Social Development and Social Risk Control Research Center Project: SR24A14.

Rural Community Governance Research Center Project: SQZL2024A01.

Agricultural Modernization and Rural Revitalization Research Center Project: AMRR2024001.

Upper Yangtze River Region Project: ZDSYS24-05.

Western Ecological Civilization Research Center Project: XBST2024-ZC003.

System Science and Enterprise Development Research Center Project: XQ23B03.

## Competing Interests

The authors declare that they have no competing interests.

## Author Contributions

- Haijun Wang conceived and designed the experiments, performed the experiments, analyzed the data, prepared figures and/or tables, authored or reviewed drafts of the article, and approved the final draft.
- Xiangdong Kong conceived and designed the experiments, performed the experiments, analyzed the data, prepared figures and/or tables, authored or reviewed drafts of the article, and approved the final draft.
- Onanong Phewnil analyzed the data, authored or reviewed drafts of the article, and approved the final draft.
- Ji Luo analyzed the data, authored or reviewed drafts of the article, and approved the final draft.
- Pengju Li analyzed the data, authored or reviewed drafts of the article, and approved the final draft.
- Xiyong Chen analyzed the data, authored or reviewed drafts of the article, and approved the final draft.

- Tianhui Xie analyzed the data, prepared figures and/or tables, authored or reviewed drafts of the article, and approved the final draft.

## Data Availability

The raw data are available in the Supplemental Files.

## Supplemental Information

Supplemental information for this article can be found online at http://dx.doi.org/10.7717/peerj.18586#supplemental-information.

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
