# Peer review of "Spatiotemporal prediction of alpine wetlands under multi-climate scenarios in the west of Sichuan, China"

_PeerJ, doi:10.7717/peerj.18586_

## Round 0.1 · original submission · Major Revisions

Wetlands are important environmental objects in today's environment under threatening environmental and climate change. The manuscript submitted by the authors on spatio-temporal forecasting of alpine wetlands in western Sichuan Province contains valuable results of a regional study. The manuscript corresponds to the direction of the journal PeerJ. After reviewing the manuscript, the reviewers expressed some comments and suggestions, which I ask you to read and respond to. Therefore, it needs to be revised in accordance with the comments.

·

Basic reporting

This is a very interesting study. This study employed a large sample and composite supervised classifcation algorithms to classify alpine wetlands and generate wetland maps, based on the Google Earth Engine cloud computing platform. Meanwhile, using the WorldClim dataset as environmental variables, we predicted the future distribution of wetlands in western Sichuan under multiple climate scenarios. The results are of scientific significance to the study of alpine wetlands. Detailed comments on the article are as follows:
(1) About the time scale of the data.
First of all, in this region, the wetland area changes significantly in different seasons. I don't understand why the author chose the image data from July and August for the study. Second, authors need to explain the time scale of the predicted results. Do the predictions in this article only represent the future months of July and August? Finally, the author should include evidence on whether the sample size of 65 images meets the requirements of the model.
(2) About environment variables.
In the current environment variable, the author uses the data of weather station for spatial interpolation to get the raster data set. However, the spatial interpolation method is not in line with the regional principle and model requirements to some extent. Because the elevation differences in this area are so significant, the authors' results using only 12 sites are not convincing. I recommend that authors use ERA5's reanalysis data product to get more accurate results.
(3) Limitations of the study
In the discussion chapter of the paper, the author mentioned the driving process of wetland area change, but the author only compared the data differences among different indicators. In practice, the driving process of wetland area change is very complicated, especially in the area with an altitude above 3000m. Therefore, the authors need to discuss and analyze the driving factors more comprehensively and clarify the limitations of the paper.
(4) About the quality of the figure
In general, the information of the map should be complete, which is conducive to the reader's reading. Figure 3. A legend of wetlands that need to be replenished. Figures 4 and 5 need to be supplemented with a serial number for each image. The slopes in Figures 7 and 8 should retain two significant decimals. In addition, the definitions of Slop, SlopMean, and Slopeupmean require remarks.

Experimental design

No comment!

Validity of the findings

No comment!

Additional comments

No comment!

·

Basic reporting

Please see review report

Experimental design

Please see review report

Validity of the findings

Please see review report

Additional comments

Please see review report

---

## Round 0.2 · accepted · Accept

The resubmitted manuscript is significantly improved and of higher quality. The research results are well structured, logically presented, and relevant. After reviewing the manuscript, the reviewers expressed a positive opinion about the significant increase in its quality compared to the previous version, as well as the readiness of the material for publication.

·

Basic reporting

The revised manuscript can explain the previous problems. Overall, I think the revised manuscript has reached the level of the journal. This is a very interesting work, and this work also gives me an inspiration.

Experimental design

No comment.

Validity of the findings

No comment.

Additional comments

No comment.

·

Basic reporting

Adequately improved based on previous review

Experimental design

Adequately improved based on previous review

Validity of the findings

Adequately improved based on previous review

Additional comments

I would like to thank the authors for addressing my concerns based on the review of the first submission. The manuscript is now ready for publication in my opinion.